

# Quantifying hazards resilience by modeling infrastructure recovery as a resource constrained project scheduling problem

Taylor Glen Johnson[1], Jorge Leandro[1], and Divine Kwaku Ahadzie[2]

[1]Research Institute for Water and Environment, University of Siegen, Paul-Bonatz-Str. 9-11, 57076 Siegen, Germany
[2]Centre for Settlements Studies, Kwame Nkrumah University of Science and Technology, Kumasi, Ghana

**Correspondence:** Taylor Glen Johnson (taylor.johnson@uni-siegen.de)

**Abstract.** Reliance on infrastructure by individuals, businesses, and institutions creates additional vulnerabilities to the disruptions posed by natural hazards. In order to assess the impacts of natural hazards on the performance of infrastructure, a framework for quantifying resilience is presented. This framework expands upon prior work in the literature to improve the comparability of the resilience metric by proposing a standardized assessment period. With recovery a central component of
assessing resilience, especially in cases of extreme hazards, we develop a recovery model based upon an application of the resource constrained project scheduling problem (RCPSP). This recovery model offers the opportunity to assess flood resilience across different events and also, theoretically, between different study areas. The resilience framework and recovery model have been applied in a case study to assess the resilience of buildings infrastructure to flooding hazards in Alajo, a neighborhood in Accra, Ghana. The results show that for the three flood events investigated (5, 50 and 500-year return periods), the 300-day
resilience of the buildings infrastructure in Alajo was quantified as 0.94, 0.82 and 0.69, respectively. In practical terms, each value reflects the ability or inability of the system to maintain its function during the reference period for the given flood event, with zero corresponding to a complete loss of function and one when unaffected. This information is valuable for identifying the vulnerabilities of buildings infrastructure, assessing the impacts resulting in reduced performance, coordinating responses to flooding events, and preparing for the subsequent recovery.

## 1 Introduction

Since the adoption of the European Floods Directive (European Commission, 2007), risk-based hazards management has become the dominant strategy for reducing the impacts of natural hazards throughout Europe and much of the world. While this strategy has proven effective at reducing the impacts, preventing the loss of life, and easing the economic burden to communities and regions following some hazard events, it generally employs a singular decision-making variable: costs resulting from
damages (Merz et al., 2010; Disse et al., 2020). On its own, damage reduction captures only a single dimension of the impact from natural hazards. Therefore, the management strategies developed from a risk-based approach are similarly limited.

In recent literature, the potential for a more evolved strategy has emerged, referred to as resilience management. This management strategy not only considers the costs associated with direct damages, but instead considers the performance of the infrastructure system over time, including through both the event and recovery phases of the hazards management cycle (Chen





and Leandro, 2019; Leandro et al., 2020). Rather than singularly seeking to reduce the damages caused by natural hazards, resilience management is focused on the ability to resist, recover and adapt to the hazard. Inbuilt in this strategy is an emphasis on maintaining a high performance of the infrastructure vital to everyday life. This paradigm shift represents the evolution of management strategies from viewing hazards as foes which must be defeated to viewing them as opportunities to adapt. While this strategy shows promise for improving upon the present management methods, it currently lacks clarity in its definition

and implementation, leading to diverging themes in the literature. This work therefore lays out a framework for implementing resilience to support hazards management by demonstrating a robust method to model recovery based on an application of the resource constrained project scheduling problem (RCPSP).

## 1.1  Defining Hazards Resilience

Before resilience can be effectively quantified, it must first be well defined. Conceptualizations of resilience as they relate to

hazards management can be categorized into three primary groups of increasingly complex interpretations of system dynamics (Disse et al., 2020): engineering resilience, ecological resilience, and social-ecological or evolutionary resilience.

Engineering resilience is a relatively simplified conceptualization which is derived from engineered systems rather than natural systems. According to this definition, the system experiences some reduction in functionality due to the hazard while at the same time resisting its effects. The system then begins to recover once the hazard subsides, returning some period of

time later to its original level of functionality. This definition assumes that the system exists in a stationary ideal state prior to the hazard event and always seeks to return to this same ideal state after the hazard subsides (Disse et al., 2020; Liao, 2012; Rodina, 2019).

The conceptualization of ecological resilience is relatively more complex than that of engineering resilience. While this definition is similar to that of engineering resilience in some regards, the primary difference is that the initial and final states

of the system are not considered ideal. Rather, the system has the ability to change state by finding a new equilibrium or "new normal" following the hazard event through adaptation (Disse et al., 2020; Liao, 2012; Rodina, 2019).

A third conceptualization of resilience is the most complex interpretation of the three. Social-ecological resilience (or evolutionary resilience) considers that the composition of a system is more dynamic than the previous definitions allow. In this interpretation, the system has no equilibrium state, but is rather in a perpetual state of change and adaptation, becoming more

or less resilient as it reacts to hazards (Disse et al., 2020; Davoudi, 2012).

While it can be tempting to conclude that because social-ecological resilience considers the most complex interpretation of system resilience, that it must be the most appropriate definition for hazards management. Indeed, numerous articles in the literature support this interpretation (Liao, 2012; Rodina, 2019). However, the selection of a definition should reflect the targeted complexity of and consequently be reflected in the uncertainty associated with the results of assessment. For example,

in cases where data scarcity exists, it can be unrealistic to apply a highly complex model with the assumption of higher accuracy. Rather, a simplified model of system interactions and individual decisions is perhaps a more appropriate model in this case with a proportionately large uncertainty included to reflect the potential inaccuracies associated with the aforementioned simplifications.




To this end, resilience is defined in this framework as the ability of a system to maintain functionality while absorbing the
effects of a flood and recovering to a state of equilibrium in a timely manner through restoration of its critical infrastructure. This interpretation is based largely upon the definition proposed by Field et al. (2012) and corresponds to an ecological conceptualization.

## 1.2 Assessing Hazards Resilience

Numerous frameworks for assessing and quantifying resilience have been presented in the literature. Distinctions can be made, however, by narrowing the scope to the resilience of the urban environment to natural hazards. In this branch of the literature, some common themes have emerged. One major commonality among the frameworks focused on the urban environment is the idea of persistent change. The panarchy model of adaptive cycle is one such illustration of the dynamics of urban resilience (Holling, 2001; Davoudi, 2012). This temporal attribute of urban systems demonstrates the necessity of assessing resilience as a time series (Chen and Leandro, 2019; Leandro et al., 2020).

It is generally accepted that resilience, being an abstract concept, cannot be directly measured, but rather must be estimated using indirect measurements via indicators of system performance (Hinkel, 2011; Schipper and Langston, 2015). A composite index of normalized indicators is generally the most commonly utilized method for achieving this goal. Through careful selection and weighting of the indicators, a proxy metric of system performance can be developed, by which an assessment of resilience can be derived (Cutter et al., 2010, 2014).

In the context of infrastructure resilience, the value of performance is assessed over some reference period, generally a period of time encompassing the effects of a disrupting event. Cimellaro et al. (2010) quantifies resilience of infrastructure as the normalized integral of the performance function over the reference period (Cimellaro et al., 2016). Due to its communicable nature and unambiguous calculation, this method for quantifying resilience provides a clear metric for assessing the benefits associated with various interventions or mitigation strategies (Cimellaro et al., 2011, 2015), a necessary attribute for operationalizing resilience in hazards management.

## 1.3 Modeling Disaster Recovery

The work by Kates and Pijawka (1977) is one of the earliest attempts to understand the post-disaster recovery process and to create a conceptual model (Miles and Chang, 2003; Miles et al., 2019). In their research, a four-stage model was presented, composed of sequential but partially overlapping stages. According to the model, disaster recovery begins with an emergency period, then a restoration period, followed by a replacement reconstruction period, and ends with a commemorative, betterment and development reconstruction period. However, this conceptualization of recovery as an orderly progression of distinct periods has been criticized, with most arguing that recovery is instead highly uncertain due to the influences of decision making and social attributes (Chang, 2010; Nejat and Damnjanovic, 2012b; Miles et al., 2019).

Since this early work, many attempts have been made to better replicate the complex interactions involved in disaster recovery through various modeling techniques. Cimellaro (2016) categorizes disaster recovery models into two broad groups: analytical and empirical. In this context, empirical recovery models are those derived from observed data or based on expert



input. According to Miles et al. (2019), machine learning has recently been utilized in the development of empirical models of housing recovery (Zhang and Peacock, 2009; Nejat and Ghosh, 2016). This approach requires sufficient empirical data for training or development, which may not be available in all cases. Analytical recovery models on the other hand, are defined as

those which have been derived from numerical simulations of system responses (Cimellaro, 2016). Agent-based models have been utilized as analytical models of housing recovery (Nejat and Damnjanovic, 2012a; Eid and El-adaway, 2017). However, agent-based models are often criticized for having a lack of transparency, being difficult to evaluate or assess, and needing to strike a balance between being overly simple or overly complicated, for which there is little consensus (Sun et al., 2016; Chen, 2012). Extensive work has been conducted in way of developing discrete-event and stochastic simulation models of recovery

(Miles and Chang, 2006, 2011; Burton et al., 2018; Miles, 2018; Burton et al., 2019; Longman and Miles, 2019). Development of these models requires a thorough understanding of the specific processes directing the system responses. As a potential alternative to the current approaches, we propose to model buildings infrastructure recovery as an application of the RCPSP due to its physically-based parameters (like availability of required resources and time needed for completing tasks) and the straightforward nature of its implementation.

The only known mention in the literature of applying the RCPSP to model disaster recovery is in the work by Miles et al. (2019). However, only the potential for applying the method for modeling lifeline infrastructure recovery is presented, for which an example from Isumi et al. (1985) is provided. To the best of our knowledge, applying the RCPSP to model infrastructure recovery in general, and specifically for modeling the recovery of buildings infrastructure, is a completely novel approach.

## 2   Methods

The following methods are divided into two distinct parts. The first, Section 2, provides a generalized description of the methods for setting up the model to simulate the recovery of infrastructure from a natural hazard. This is intended to demonstrate the broad applicability of the model to a wider context than any singular case study would otherwise allow. The second part, Section 3, demonstrates the specific methods used to develop a model for a case study of the recovery of buildings following flooding in Accra, Ghana.

### 2.1   Quantifying Hazards Resilience

The quantification of infrastructural resilience requires a means for assessing the performance of the system on a component-level basis. These component-level measurements serve as proxy indicators of the overall system function. Indicators $X$ are normalized according to Eq. 1, where $x(t)$ is a measurement indicating the performance of a particular component of infrastructure at time $t$. The values $x_{min}$ and $x_{max}$ correspond to the minimum and maximum values of the measured variable,

respectively. The minimum and maximum values can refer to either the range which is possible for the value, or rather the range which is considered acceptable. Additionally, indicators which increase with improved system performance (positively correlated) are deemed positive indicators and indicators which decrease with improved system performance (negatively correlated)





are deemed negative indicators. The two types of indicators are normalized differently so to produce a positive correlation with system performance (Cutter et al., 2010, 2014; Scherzer et al., 2019).

$$X(t) = \begin{cases} \dfrac{x(t) - x_{min}}{x_{max} - x_{min}}, & \text{positive;} \\[2ex] \dfrac{x_{max} - x(t)}{x_{max} - x_{min}}, & \text{negative.} \end{cases} \qquad X \in [0,1] \tag{1}$$

In order to assess the system performance as a whole, the individual indicators are combined using a composite index, as in Eq. 2, where $w_i$ is a weighting factor applied to indicator $X_i$. The magnitude of the weighting factor reflects the relative importance of the indicator (Cutter et al., 2010, 2014; Scherzer et al., 2019).

$$P(t) = \frac{\sum\limits_{i=1}^{n} X_i(t) \cdot w_i}{\sum\limits_{i=1}^{n} w_i} \tag{2}$$

Hazard models are used to simulate the impacts to infrastructure for a particular scenario or event $e$. The performance $P(t)$ is assessed over a period of time encompassing the influences of the hazard on the system. This time interval is referred to as the assessment period $\Delta t_a$ and begins at the onset of the hazard event $t_{e_i}$. The resilience $Re$ of the infrastructure to hazard event $e$ is then quantified as the normalized integral of the performance over the assessment period (Cimellaro et al., 2010, 2016). Eq. 3 presents a mathematical formulation of the described framework.

$$Re = \frac{1}{\Delta t_a} \cdot \int\limits_{t_{e_i}}^{t_{e_i} + \Delta t_a} P(t) \cdot dt \tag{3}$$

While constructing a thorough measure of $P(t)$ through careful selection of indicators and their respective weights is important for accurately describing the performance of the infrastructure system (Cutter et al., 2010, 2014), systematic selection of the assessment period $\Delta t_a$ is just as important. Inspection of Eq. 3 demonstrates that the resulting value of $Re$ is strongly dependent on this parameter. The following sections present a framework for determining $\Delta t_a$.

**2.1.1   Assessment Period**

The timeline of the direct influences of the hazard event on the infrastructure system can be broken into two periods conceptually: an event phase and a recovery phase (Chen and Leandro, 2019; Leandro et al., 2020). The event phase is characterized by





the physical impacts of the hazard. This phase is enveloped by the time at which the event begins, $t_{e_i}$, until the direct effects of the hazard have subsided $t_{e_f}$. The event phase interval $\Delta t_e$ is therefore formalized by Eq. 4.

$$\Delta t_e = \left[ t_{e_i}, t_{e_f} \right] \tag{4}$$

Immediately following the event phase, recovery in various capacities can begin. The recovery phase encompasses the period from subsidence of the hazard $t_{e_f}$ until a steady state is achieved $t_s$, provided that the dynamics of the system are conceptualized using either an engineering or ecological definition of resilience. Activities supporting a return to a high quality of performance and all potential adaptations to the system are contained in this phase. When assessing the resilience of the system to multiple

hazards, the equilibrium time is taken as the maximum of the recovery times $t_{s_{max}}$, which generally corresponds to the scenario with the largest magnitude hazard. The recovery phase interval $\Delta t_r$ is therefore formalized by Eq. 5.

$$\Delta t_r = \left( t_{e_f}, t_{s_{max}} \right] \tag{5}$$

According to the definitions presented, the assessment period $\Delta t_a$ encompasses both the event and recovery phases. Therefore, the interval can be formalized by Eq. 6.

$$\Delta t_a = \Delta t_e + \Delta t_r = \left[ t_{e_i}, t_{s_{max}} \right] \tag{6}$$

While these definitions of the time parameters are accepted in the literature, the specifics of this approach present an issue for comparability. Consider that as the recovery time is reduced, the assessment interval is likewise reduced by an equal amount of time. Reduction in recovery time is, by definition, indicative of an increase in system resilience. However, the resulting value of resilience according to these equations does not react proportionally.

Consider also, that two different systems might recover at dramatically different rates from the same hazard scenario. According to the current convention, it is possible that both systems are evaluated as being equally resilient. As achieving a timely recovery is likewise indicative of higher system resilience, the metric should rather produce different results for these two systems. Therefore, an alternative approach for selection of the time parameters is proposed.

### 2.1.2   Standardized Assessment Period

Rather than utilizing the maximum recovery time $t_{s_{max}}$, which is specific to each system and dependent on the set of hazard scenarios applied in the investigation, we propose to standardize the assessment interval. In practice, this alternative method requires selection of an appropriate constant interval which envelopes the target responses of the system. The updated assessment period is given by Eq. 7.



$$\Delta t_a = \text{const.} \tag{7}$$

This alternative approach allows for direct quantitative comparison of the resilience across systems and hazards, for assessments utilizing the same $\Delta t_a$. The interval applied in the investigation is to be communicated alongside the value of resilience for clarity. For example, the system resilience quantified using an assessment period of 100-days is reported as "100-day resilience" or $Re_{\text{100-day}}$.

    It can be deduced that an assessment utilizing a reference period significantly larger than the recovery time has the effect

of increasing the magnitude of the calculated value and reducing the sensitivity of the resilience metric. Therefore, it can be necessary to apply different assessment periods between studies, depending on the sensitivity required. However, it remains necessary to use the same assessment period for two studies in order to compare them. It is for this reason that we propose reporting the assessment period alongside the metric for clarity.

### 2.1.3   Assessment Period for Extreme Events

Extensive impacts to infrastructure can occur either during extreme events or even during moderate events if the system is highly vulnerable. Longer recovery times are generally expected in cases with extensive impacts. In the case that the recovery phase is much longer than the event phase, measurement of the reduction in performance over the event phase is largely insignificant and therefore unnecessary if quantification of resilience is the primary objective of the investigation. Thereby, it is hypothesized that resilience to extreme hazards can be estimated with a similar accuracy by assessing system performance

over the recovery phase only. Therefore, Eq. 3 can be modified to reflect this change, resulting in Eq. 8.

$$Re \approx \frac{1}{\Delta t_a} \cdot \int_{t_{e_f}}^{t_{e_f}+\Delta t_a} P(t) \cdot dt \qquad \text{if } \Delta t_e << \Delta t_r. \tag{8}$$

### 2.2   Infrastructure Recovery Model

Following the proposal of an assessment period for extreme events in which it is hypothesized that the event phase is largely insignificant for the quantification of resilience, the emphasis is rather placed on the recovery phase. Therefore, it is necessary

to develop a model of infrastructure recovery which can be used to measure the performance of the system over the assessment period. The following section outlines the general case of the model.



**Table 1.** Database of recovery scenarios ($R$) resulting from the discretization of damages ($D_i$) and the classification of structures ($S_i$).

|  | $S_i$ | $S_{i+1}$ | $\cdots$ | $S_n$ |
|---|---|---|---|---|
| $D_i$ | $R(S_i, D_i)$ | $R(S_{i+1}, D_i)$ | $\cdots$ | $R(S_n, D_i)$ |
| $D_{i+1}$ | $R(S_i, D_{i+1})$ | $R(S_{i+1}, D_{i+1})$ | $\cdots$ | $R(S_n, D_{i+1})$ |
| $\vdots$ | $\vdots$ | $\vdots$ | $\ddots$ | $\vdots$ |
| $D_n$ | $R(S_i, D_n)$ | $R(S_{i+1}, D_n)$ | $\cdots$ | $R(S_n, D_n)$ |

### 2.2.1 Infrastructure Recovery Concept and Assumptions

In the framework of this model, infrastructure recovery is conceptualized as a project having a beginning and an end, which is composed of a collection of smaller tasks. Provided that the damages to the infrastructure resulting from a hazard are either known or well estimated, the individual tasks of the larger project can be inferred.

The individual components of an infrastructure system can be categorized into structural classes according to a predetermined criteria, so to capture the range of required recovery pathways. Applying prior knowledge of the susceptibility of the various infrastructure components, the potential damage states due to a hazard acting on each component can be categorized or discretized. These damage classes can then be used to determine which actions or tasks must be carried out in order to recover the component, as well as how the tasks relate to one another. We will refer to the collection of tasks corresponding to a single component and a single damage class as a recovery scenario. Upon thorough investigation of the components, the produced recovery scenarios are collected and stored into a database. An example of this database is shown in Table 1.

When the system is subjected to a hazard, a damage assessment is carried out on a component-level basis. Two important pieces of information about each component can then be identified: the structural class to which the component belongs and the damage class which has resulted from the effects of the hazard. Using these two classifications, we can then collect the corresponding recovery scenario from the database. After all components have been assessed in this way, the collected recovery scenarios are grouped together according to the precedence relationships between tasks to form the larger project, referred to as a recovery plan. At this point, all tasks which must be completed in order to recover the infrastructure system are known. The processing time of each task must then be determined by the recovery model. This model is primarily based upon the following four assumptions regarding the recovery process.

**Assumption 1.** It is assumed that recovery is prolonged by the time required to complete each individual task of the overall process. For example, for a given structure, there are many tasks which must be completed in order to recover the structure to its pre-hazard condition. Each of the tasks will require some finite amount of time to complete. While it may be possible that the duration is either longer or shorter due to a variety of factors, the duration of each individual task remains an important factor in the overall expedience of recovery.

**Assumption 2.** Just as the duration of each task affects the rate of recovery, so too is the effect compounded when considering the possibility that one or more tasks may not begin until one or more preceding tasks have ended. Although some tasks can



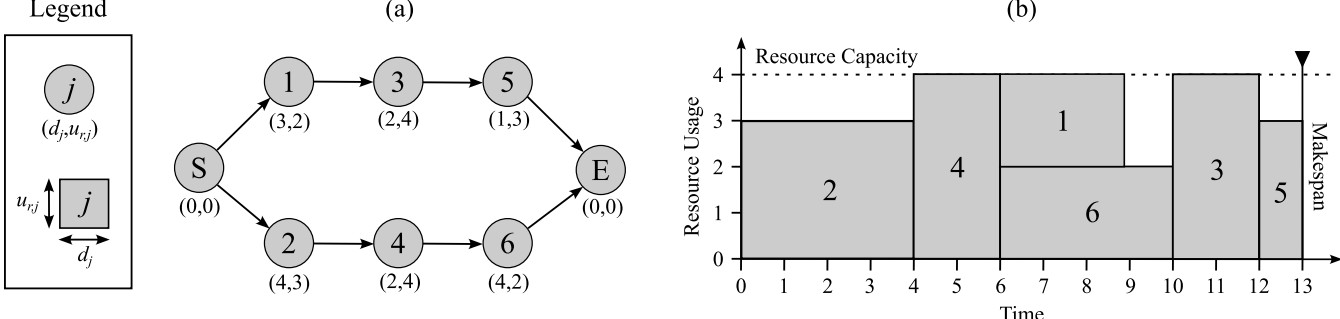

**Figure 1.** (a) A basic example of an RCPSP represented by an AON directed graph and having only a single constraining resource $r$. Each node of the graph is a task or job $j$ with a corresponding duration $d_j$ and resource requirement $u_{r,j}$. Arcs represent the precedence relationship between jobs. (b) A solution to the presented problem when the capacity of resource $r$ is 4 and makespan minimization is the objective. Adapted from Kolisch and Hartmann (1999).

be conducted in parallel with other tasks, it is also possible that one or more tasks must first be completed in order to begin the next step in a recovery process. Therefore the precedence relationships between tasks affect the expedience of recovery.

**Assumption 3.** Critical resources can be defined as the items or services required to carry out the tasks of the recovery. It can be assumed that there exists only a finite amount of each critical resource which is available to the tasks at any given time. This limited quantity is the resource's capacity. If there is not enough of a resource, or the resource capacity is less than the demand, then tasks must be delayed until the necessary resources become available. Therefore, each resource capacity is also a potential limiting factor for recovery.

**Assumption 4.** The final assumption is that recovery is, to some degree, naturally optimized. This optimization is due to the cumulative result of each affected entity seeking a speedy recovery for itself. For example, if a component of infrastructure is damaged by a hazard and the repairs are considered feasible, then there will be a desire for the recovery of the component to be carried out in a timely manner.

### 2.2.2 Resource Constrained Project Scheduling Problem

The four model assumptions simplify the problem of infrastructural recovery, creating the opportunity to formulate the problem as an application of the RCPSP. The base case of the RCPSP consists of tasks (the individual, discrete components of a larger project) which should be optimally scheduled according to the objective of makespan minimization. This problem is generally illustrated using activity-on-node (AON) directed graphs, where each node represents a task and each arc represents a precedence relationship between two tasks. Two additional nodes having no duration are appended to the start and end to

demarcate the beginning and completion of a project. Resources are treated as renewable in the base case. That is, each resource has a fixed capacity which is renewed at each time interval. Unused resources from a previous time interval do not carry over to the next. An example of a simple RCPSP is shown in Fig. 1.

This base case of the RCPSP is of the NP-hard variety (Kolisch and Hartmann, 1999). Therefore, an optimal solution can generally be found in a reasonable time when the search space is limited to relatively few tasks and constraining resources.



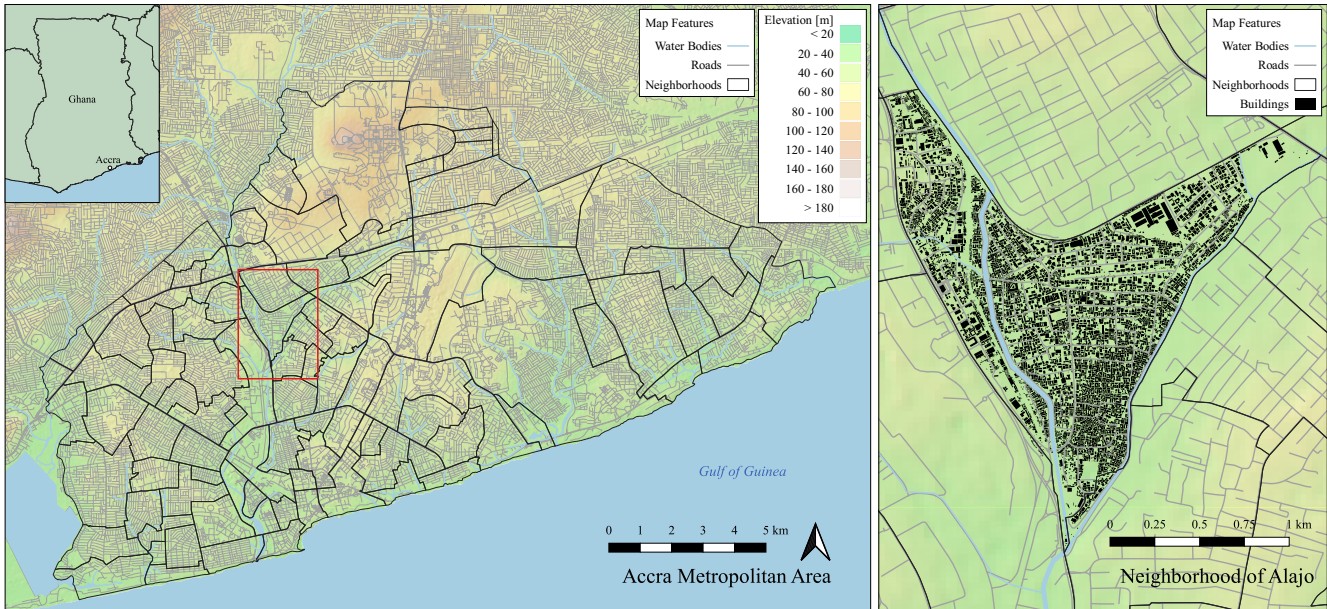

**Figure 2.** *Left*: Map showing the location of the neighborhood Alajo within the greater Accra Metropolitan Area (Engstrom et al., 2013). *Right*: An enlarged map of the study area including the buildings considered in the model. Adapted from OpenStreetMap data acquired from Geofabrik GmbH, © OpenStreetMap contributors 2020. Distributed under the Open Data Commons Open Database License (ODbL) v1.0.

However, when scheduling many tasks or including many resource constraints, the problem grows larger and requires much more computational time to solve. In this case, finding an optimal solution can become infeasible within a reasonable solve time and a heuristic algorithm must be implemented to approximate the solution.

## 3 Case Study: Recovery of Buildings from Flooding in Accra, Ghana

In order to demonstrate an application of the described framework, the recovery model was tested in a case study of Alajo,
a district within the greater Accra Metropolitan Area, Ghana (Fig. 2). Alajo is of particular interest because it is composed of a dense mix of building types and is situated at the confluence of two major storm-water drainage canals. This district is thereby prone to flooding hazards. Further compounding the issue of flooding, much of the floodplain is occupied by informal residential development, making the consequences of flooding particularly severe.

### 3.1 Quantifying Flood Resilience of Buildings Infrastructure

This case study is focused solely on the recovery of buildings following flooding events. Therefore, only an indicator of the state of buildings is necessary for quantifying resilience in this scope. The state of each building $b$ is considered from the point of view of the occupant of the building. Thereby, a building is considered to exist in a binomial state at any time $t$: either occupied (1) or unoccupied (0). In the occupied state, the building can currently be used by its occupant for its intended





purpose, whether for shelter in a residential building or for economic activity in a commercial building. In the unoccupied
state, a building is damaged to an extent that it cannot be used for its intended function and must be repaired before a return to
occupancy is possible. The indicator is provided in Eq. 9.

$$X_b(t) = \begin{cases} 0, & \text{if } b \text{ unoccupied at } t; \\ 1, & \text{if } b \text{ occupied at } t. \end{cases} \tag{9}$$

The relative importance of each building $b$ is determined by its footprint area $A_b$. It is assumed that recovering a building with
a larger footprint area indicates a greater increase in performance than a building with a smaller area. The final performance
metric is then calculated as the total area of buildings which are occupied at time $t$ versus the total area of all buildings in the
set $B$. Eq. 2 therefore becomes Eq. 10.

$$P(t) = \frac{\sum\limits_{b \in B} X_b(t) \cdot A_b}{\sum\limits_{b \in B} A_b} \tag{10}$$

The event phase for the flooding hazard is conceptualized as the onset of inundation at $t_{e_i}$ until the flood waters recede and
inundation ends at $t_{e_f}$. Due to the severity of prior flood damages in Alajo, the extreme event assessment period is applied
for quantifying resilience in this case study. Therefore, the event phase is neglected. We will also utilize a relatively large
assessment period of 300 days by setting $\Delta t_a = 300$ in Eq. 8, to capture the lengthy duration of the recovery.

### 3.2 Buildings Infrastructure Recovery Model

In the recovery model, recovery plans take the place of projects in the RCPSP. A recovery plan includes all of the tasks which
must be completed by individuals in order to bring their building to a safe and functional state following damages from a flood.
First we assess each building to determine into which structural classification and which damage class it falls. Knowledge
of these two classifications allow us to infer the tasks which must be completed to recover that particular building. The set of
tasks corresponding to a single structural class and damage class is a recovery scenario. We collect all of the identified recovery
scenarios into a database, provided in Table 2.

Upon assessment of the structural type and damage state of all buildings following a hazard event, the corresponding recovery
scenarios are collected and grouped together to form the recovery plan. In the case of the buildings in Alajo, it is assumed that
there is no precedence relationship between buildings. That is, the processing of any task is not directly dependent on the
processing status of the tasks of other buildings, only on the precedence relationships between other tasks of the same recovery
scenario and the availability of critical resources. Therefore, all recovery scenarios are placed in parallel with one another in
the recovery plan, as shown in Fig. 3. The resulting recovery plan is then solved using optimization.



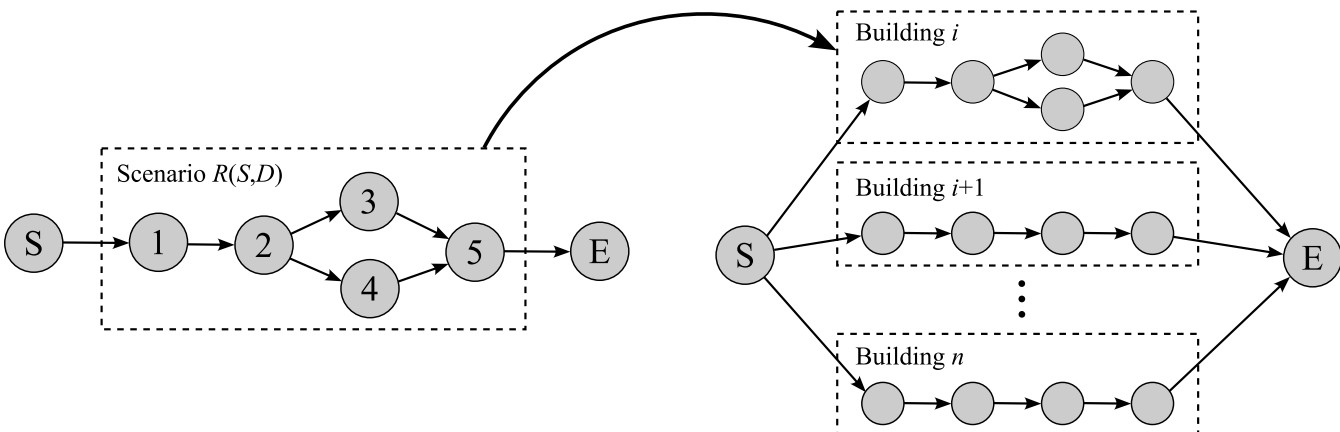

**Figure 3.** *Left*: An example of a single recovery scenario $R$ corresponding to structure class $S$ and damage state $D$. *Right*: A recovery plan for a single hazard event with all buildings in parallel.

**Table 2.** Recovery scenarios resulting from the discretization of damages and the categorization of buildings.

|  | IR: Informal Residential | FR: Formal Residential | CI: Commercial & Industrial |
| --- | --- | --- | --- |
| D0: Insignificant | IR-D0 | FR-D0 | CI-D0 |
| D1: Moderate | IR-D1 | FR-D1 | CI-D1 |
| D2: Heavy | IR-D2 | FR-D2 | CI-D2 |
| D3: Complete | IR-D3 | FR-D3 | CI-D3 |

While many algorithms already exist which provide an exact solution to the mathematical formulation of the RCPSP in small scale, expanding the search space makes finding an exact solution potentially infeasible (Kolisch and Hartmann, 1999). Therefore, approximating this problem on a larger scale required an alternative optimization method. A review of the literature pointed to a few capable heuristic strategies. Among the best performers are simulated annealing, tabu search and genetic algorithms (Kolisch and Hartmann, 1999). Ultimately, the genetic algorithm described by Hartmann (2002) was chosen due to its benchmark performance.

### 3.3 Building Classes

Three classes of buildings were established in order to capture both the vulnerability of the structure itself, as well as the likely economic capability of the building's occupants. These are informal residential, formal residential, and commercial and industrial. Each of the building types were classified according to four indicators, adapted from the methods used by the World Bank (2017) to classify buildings in Accra. The first indicator is a quantitative assessment of the footprint area of each building. The next indicator is the quantitative measurement of the density of neighboring buildings. Both of these quantitative measurements were calculated using a shapefile of the building footprints adapted from OpenStreetMap data acquired from Geofabrik GmbH. The third and fourth indicators are the apparent quality of the roofing material and the apparent use of the





surrounding property near each building. These qualitative indicators were assessed using visual inspection of Google Earth
imagery, an example of which is provided in Fig. 5.

Informal residential buildings are likely built without approval of the responsible authorities. Therefore they may not follow
established construction practices and are generally constructed of low-quality materials. These buildings are also likely to be
located in areas where official approval would not normally be granted (flood plains, for example). It is assumed that residents
of these buildings have limited financial means to secure quality building materials when repairing damages from a flood. They
are also less able to afford the costs of skilled workers to aid in the repairs, instead relying on either unskilled workers or
their own abilities. Informal residential buildings are generally among the smallest and most dense of the three building types
considered in this study. Footprint areas of less than 100 m$^2$ and building densities of greater than 20 buildings in a 50-m radius
were considered indicative of this class. Viewed from satellite imagery, these buildings have characteristic patchwork roofs
composed of heterogeneous, low-quality materials with little or no green space in the immediate vicinity.

Formal residential buildings are considered more likely to be built with government approval following established construc-
tion methods and composed of high-quality building materials. Because there are higher costs associated with the approval and
construction of formal residential buildings, it is assumed that residents of these buildings likewise have greater financial means
available for repairs or reconstruction following flood damages, lending to a relatively expedient recovery. Formal residential
buildings fall into a middle range of footprint areas (100 to 300 m$^2$) and building densities (10 to 20 buildings in a 50-m ra-
dius). Views of quality homogeneous roofing materials, driveways, and green yards surrounded by walls or fences are typically
visible in satellite images of these buildings.

Commercial and industrial buildings serve the purpose of providing space for economic activity. Such buildings are typically
built of robust construction and high-quality materials. Because of the economic nature of these buildings, it is generally
assumed that owners of commercial and industrial buildings have a greater financial means for buying materials and hiring
skilled labors to repair damages due to floods. These buildings are generally larger in size than the other two building types
and utilize large spaces for business activities. A footprint area or greater than 300 m$^2$ and a building density of less than 10
buildings in a 50-m radius are indicative of commercial and industrial buildings. Satellite imagery often reveals machinery and
material storage in the area surrounding the building. The indicators and a description of their values for each building class
are outlined in Table 3.

**3.4    Damage Classes**

Damages to the buildings were determined using a simple correlation between the inundation water depth $h_w$ and a vulnera-
bility function specific to the building class. The generalized version of the vulnerability function is given by Eq. 11.

$$d(h_w) = -c \cdot \exp\left[-k \cdot \frac{h_w}{h_b}\right] + c \qquad d \in [0, d_{max}] \tag{11}$$





**Table 3.** Description of indicators used for classifying buildings. Adapted from World Bank (2017).

| Indicator | Building Class | | |
| --- | --- | --- | --- |
| | IR: Informal Residential | FR: Formal Residential | CI: Commercial & Industrial |
| Building Size | <100 m$^2$ | 100-300 m$^2$ | >300 m$^2$ |
| Building Density | >20 buildings in 50-m radius | 10-20 buildings in 50-m radius | <10 buildings in 50-m radius |
| Roof Material | heterogeneous, low-quality | homogeneous, high-quality | homogeneous, high-quality |
| Property Use | typically very little surrounding property; little to no green space | open yards and driveways; often surrounded by walls or fences | vast open space; often used for storage of commercial goods; can be paved for driving and parking vehicles or machinery |

**Table 4.** Damage curve parameters corresponding to each building class. Derived from regression of the work presented in Englhardt et al. (2019).

| Building Class | $c$ | $k$ | $d_{max}$ |
| --- | --- | --- | --- |
| IR: Informal Residential | 1.074 | 2.516 | 1.00 |
| FR: Formal Residential | 0.856 | 2.918 | 0.81 |
| CI: Commercial & Industrial | 0.694 | 2.793 | 0.65 |

The height of the flood water above the base of the building $h_w$ is determined by taking the mean water depth within
the area of the building's footprint. The height of the building $h_b$ is assumed to be 2.5 meters for all buildings simply due
to insufficient data indicating otherwise. Each building is assumed to have a maximum possible damage $d_{max}$ based on the
potential to reuse aspects of the structure even after complete inundation. Finally, $c$ and $k$ are constants relating the materials
used to construct each building with vulnerability to flood damage. The constants corresponding to each building class were
derived from regression of the work presented in Englhardt et al. (2019) and are presented in Table 4.

The damage function was then discretized into four finite categories based generally on the tasks which are required to
recover the building at each level of damage. The damage classes are summarized in Table 5. Insignificant damage corresponds
to the very lowest damage level. At this damage level, it is assumed that either nothing needs to be done to repair the building, or
the damages are only cosmetic. Therefore, any cosmetic repairs are neglected from the recovery plan and the building remains
occupied from the start of the recovery. The damage factor associated with insignificant damage is in the range 0.00 to less
than 0.01.

The next damage level is moderate damage. Buildings experiencing moderate damage require structural repairs. However,
the repairs are simple enough that they can be carried out in a relatively short time and without a significant amount of human
and material resources. The range of damage factors categorized as moderate is from 0.01 to less than 0.30.





**Table 5.** The discretization of damages into finite classes as applied in the model. Adapted from Bai et al. (2009) and Kreibich et al. (2009).

| Damage Class | Damage Factor | Description |
| --- | --- | --- |
| D0: Insignificant | $0.00 \leq d < 0.01$ | Unaffected or requiring only cosmetic repair. |
| D1: Moderate | $0.01 \leq d < 0.30$ | Repairable structural damage has occurred. |
| D2: Heavy | $0.30 \leq d < 0.80$ | Structural damage requires major repairs. |
| D3: Complete | $0.80 \leq d \leq 1.00$ | Extensive damage. Repair of most elements not feasible. |

Heavy damage is characterized by major structural damage. Repairs in this category require extensive work, some of which requires skilled labor. There is a high requirement for critical human and material resources in order to bring buildings with heavy damage back to a state of occupancy. The damage factor range for this category is from 0.30 to less than 0.80.

The final damage level is complete damage. In this class, buildings will have experienced extensive damages to the point that repair of most of the building elements are no longer feasible. Therefore, the building must be demolished and a new structure built in its place. Because of the heavy financial burden and the long delay associated with reconstruction, these buildings are considered less likely to attempt repair than in other classes. The damage factor range for this category is from 0.80 to 1.00.

### 3.5 Parameter Uncertainty

A Beta-PERT distribution was utilized to model the uncertainty associated with the model parameters. This distribution was chosen for this purpose because it offers a method for translating expert input into a probability density function based on three parameters: a minimum possible value ($\alpha$), a maximum possible value ($\beta$), and a most-likely value ($m$). The Beta-PERT distribution was applied to the capacities of each of the resources and to the duration of each of the tasks.

### 3.6 Task Durations and Resource Requirements

The resources deemed critical to recovery in Alajo can be broken into two general categories: basic materials and human capital. Among the basic materials are cement, waterproof cement, sandcrete blocks, steel reinforcement bar, lumber, bitumen, and epoxy paint. The human capital critical to recovery includes common or unskilled workers, construction workers, and utilities workers. All resources are treated as renewable resources in the model. Therefore, there is a renewed daily capacity for each.

Durations of tasks were estimated in units of "time per area" to represent the increase in duration with increasing building size. This time scaling was assumed to be linear. Because a large portion of the buildings falls under the 100 m² size, this was chosen to be the reference value by which estimates would be made. As duration changes, the amount of resources required during each day of the task remains the same. Therefore, as the building size increases, the duration increases and so does the resource requirement. This is the desired effect because the assumption is that a larger building will require more time and resource to be repaired.





**Table 6.** Description of the resources applied in the model and their capacities. Units for capacity are "unit/day/100 buildings" and the provided parameters ($\alpha$, $m$, $\beta$) correspond to a Beta-PERT distribution.

| No. | Description | Unit | Capacity $\alpha$ | $m$ | $\beta$ |
|---|---|---|---|---|---|
| R1 | Cement | 50 kg | 50 | 70 | 100 |
| R2 | Sandcrete Block | piece | 50 | 70 | 100 |
| R3 | Steel Reinforcement Bar | piece | 50 | 70 | 100 |
| R4 | Construction Worker | daily wage/person | 10 | 15 | 30 |
| R5 | Utilities Worker | daily wage/person | 10 | 15 | 30 |
| R6 | Common Worker | daily wage/person | 20 | 30 | 50 |
| R7 | Waterproof Cement | 1 kg | 20 | 30 | 50 |
| R8 | Epoxy Paint | 5 liters | 20 | 30 | 50 |
| R9 | Bitumen | drum (220 liters) | 10 | 15 | 30 |
| R10 | Lumber | piece | 50 | 70 | 100 |

**Table 7.** An example of the task parameters for the Formal Residential, Complete Damage (FR-D3) scenario. Units of duration are "days/100 m$^2$" and the provided parameters ($\alpha$, $m$, $\beta$) correspond to a Beta-PERT distribution. Units of resource requirements are "unit/day" according to the unit associated with the particular resource.

| No. | Description | Duration $\alpha$ | $m$ | $\beta$ | Resource Requirements R1 | R2 | R3 | R4 | R5 | R6 | R7 | R8 | R9 | R10 |
|---|---|---|---|---|---|---|---|---|---|---|---|---|---|---|
| T1 | Seek financial assistance. | 5 | 10 | 20 | 0 | 0 | 0 | 0 | 0 | 0 | 0 | 0 | 0 | 0 |
| T2 | Professionally demolish existing structure. | 2 | 5 | 10 | 0 | 0 | 0 | 0 | 0 | 3 | 0 | 0 | 0 | 0 |
| T3 | Professionally prepare and lay new foundation. | 5 | 8 | 15 | 20 | 0 | 10 | 1 | 0 | 3 | 5 | 0 | 0 | 0 |
| T4 | Professionally build walls. | 8 | 10 | 20 | 10 | 20 | 10 | 2 | 0 | 2 | 5 | 0 | 0 | 0 |
| T5 | Professionally build roof. | 8 | 10 | 20 | 0 | 0 | 0 | 2 | 0 | 2 | 0 | 0 | 2 | 10 |
| T6 | Professionally install plumbing. | 5 | 8 | 15 | 0 | 0 | 0 | 0 | 2 | 0 | 0 | 0 | 0 | 0 |
| T7 | Professionally install electrical. | 5 | 8 | 15 | 0 | 0 | 0 | 0 | 2 | 0 | 0 | 0 | 0 | 0 |
| T8 | Professionally finish interior. | 8 | 10 | 20 | 0 | 0 | 0 | 1 | 0 | 2 | 0 | 2 | 0 | 4 |
| T9 | Reoccupy structure. | 0 | 0 | 0 | 0 | 0 | 0 | 0 | 0 | 0 | 0 | 0 | 0 | 0 |

## 3.7 Building Abandonment

In order to incorporate individual decision analysis into the proposed recovery model, an additional parameter termed the probability of building abandonment was included, allowing each of the building owners to decide not to seek to recover the structure. A probability was assigned to each of the nine recovery scenarios. Generally, the probability of building abandonment was assumed to increase with higher damage extents. Alternatively, the probability was assumed to decrease along with the assumption of available monetary resources. For example, formal residential buildings are assumed to have greater financial capacity than informal residential buildings. Therefore, the probability of abandonment is lower for formal residential buildings





**Table 8.** Probability of building abandonment for each of the recovery scenarios presented in Table 2 resulting from the discretization of damages and the categorization of buildings.

|  | IR: Informal Residential | FR: Formal Residential | CI: Commercial & Industrial |
|---|---|---|---|
| D0: Insignificant | 0.00 | 0.00 | 0.00 |
| D1: Moderate | 0.05 | 0.02 | 0.01 |
| D2: Heavy | 0.20 | 0.10 | 0.05 |
| D3: Complete | 0.50 | 0.20 | 0.10 |

than informal residential buildings for same damage level. Table 8 provides the probability of abandonment for each of the recovery scenarios.

### 3.8 Stratified Random Sampling and Monte Carlo Simulation

If each of the recovery scenarios were to involve five to ten individual tasks, as initial investigation suggested, then recovery plans would be made up of approximately 50,000-100,000 tasks given that there are approximately 10,000 buildings in Alajo. Such a large sample space would not be feasible for the optimization algorithm to approximate in a reasonable amount of computational time. Therefore, it was necessary to take smaller, random samples of buildings and apply these to the model. To this end, a stratified random sampling scheme was devised. For each flooding event assessed, buildings were divided into groups according to their classification into the nine recovery scenarios. A target sample size was randomly selected from among the groups according to the relative size of each group to the whole population of buildings. A Monte-Carlo Simulation (MCS) was then conducted with a new stratified random sample drawn at each iteration and replacement of drawn samples back to the population.

### 3.9 Flood Inundation

The Parallel Diffusive Wave Model (P-DWave) was applied to model flood inundation (Leandro et al., 2014). This model applies a first-order finite volume explicit discretization scheme on a regular grid to solve the diffusive form of the 2-D shallow water equations, as shown in Eq. 12, where $g$ is the acceleration due to gravity, $h$ is the water depth, $z$ is the bed elevation, $u$ is the depth-averaged flow velocity vector, $v_t$ is the turbulent eddy viscosity, $R$ is the source-sink term relating to rainfall or inflow, and $S_f$ is the bed friction factor. This is accomplished by neglecting all forces in the momentum equations except the gravity term and bed friction, resulting in the simplified momentum equation given by Eq. 13.

$$\frac{dh}{dt} + \nabla(uh) = R \tag{12}$$



$$g\nabla\left(h+z\right)=gS_f \tag{13}$$

The water-level surface gradient vector term is given by Eq. 14, where $S_{wx}$ and $S_{wy}$ are the water-level surface components in the x- and y-directions, respectively.

$$\nabla\left(h+z\right)=\begin{bmatrix}S_{wx}\\S_{wy}\end{bmatrix}=\begin{bmatrix}\dfrac{d\left(h+z\right)}{dx}\\\dfrac{d\left(h+z\right)}{dy}\end{bmatrix} \tag{14}$$

Manning's formula, shown in Eq. 15, is used to approximate the bed friction $S_f$, where $n$ is the Manning's roughness coefficient, $u_x$ and $u_y$ are the velocity components in the x- and y-directions, respectively, and $|u|$ is the modulus of the depth-averaged flow velocity vector, given by Eq. 16.

$$S_f=\begin{bmatrix}S_{fx}\\S_{fy}\end{bmatrix}=\begin{bmatrix}\dfrac{n^2\left|u\right|u_x}{h^{\frac{4}{3}}}\\\dfrac{n^2\left|u\right|u_y}{h^{\frac{4}{3}}}\end{bmatrix} \tag{15}$$

$$\left|u\right|=\frac{h^{\frac{2}{3}}\left(S_{wx}^2+S_{wy}^2\right)^{\frac{1}{4}}}{n} \tag{16}$$

As input, P-DWave minimally requires an elevation raster, a surface roughness raster, a rainfall hyetograph, and initial and boundary condition rasters. For the case study, a digital elevation model (DEM) was sourced from the Advanced Land Observing Satellite (ALOS) mission of the Japanese Aerospace Exploration Agency (JAXA). The DEM was altered for use in the simulation by removing sinks and other anomalies and burning-in waterway and street networks. The surface roughness raster was produced by classifying land use into six categories based on data from the World Bank (2017) and OpenStreetMap. The land use classes and their corresponding Manning's $n$ roughness coefficients were: informal residential (0.30 s/m$^{1/3}$), formal residential (0.20 s/m$^{1/3}$), industrial (0.15 s/m$^{1/3}$), natural (0.05 s/m$^{1/3}$), roads (0.03 s/m$^{1/3}$), and waterways (0.02 s/m$^{1/3}$). No inflow boundary conditions were necessary because the entire watershed was applied in the simulation. Likewise, no initial water depths were applied, rather the rainfall duration was extended in order to allow for filling of drainage canals. A triangular design rainfall was applied in the model with a storm advancement coefficient of 0.4 (unitless). The rainfall durations and intensities were sampled from intensity-duration-frequency (IDF) curves derived from historical rainfall data sourced from the Ghana Meteorological Agency (GMet). Infiltration was modeled using the SCS Curve Number method as described in





Technical Report 55 (TR-55) from USDA-NRCS (1986). Curve numbers are determined by land use and hydrologic soil groups (HSG). HSG are primarily associated with infiltration rates and textures of soils. According to the UN-FAO's Digital Soil Map of the World, the watershed is composed primarily of two soil types, Ferric Acrisols and Chromic Vertisols, both of which correspond to HSG B due to their drainage properties (Batjes, 1997). For each of the land use classes used to build

the roughness raster, a corresponding low, high and mean curve number was assigned from the tables in the TR-55 manual according to HSG B. An area-weighted composite of the curve numbers was calculated for each case to produce a low, high, and mean composite of 69, 72, and 75 (unitless), respectively. The mean composite curve number of 72 was used to calculate the excess rainfall hyetographs applied in P-DWave.

## 4   Results and Discussion

### 4.1   Recovery Model and Resilience Quantification

For each of the hazard scenarios investigated (5, 50, and 500-year floods), the recovery model was applied using an MCS with 500 iterations, drawing a stratified random sample of 100 buildings at each iteration. For the optimization, the genetic algorithm utilized a population size of 50, a probability of gene mutation of 5% ($P = 0.05$), and was limited to a maximum of 10 generations. The model produced a database for each scenario containing the scheduled start and end times for each task in

the recovery plan for every MCS iteration. It is from these databases that the following results are derived.

The scheduled tasks produced by the recovery model can be used to determine the building states (either occupied or unoccupied) at time $t$ over the assessment period. Combining this information with the known footprint areas of each building, Eq. 10 can be applied to produce a timeline of the performance of the buildings infrastructure. From the MCS, the mean and 95% confidence intervals of the performance curves were derived. The resulting performance curves for the three investigated

scenarios are shown in Fig. 4. It is apparent from the figure that the performance is generally reduced as the return period of the scenario increases. One can also see that the shape of the curve becomes more flat with the increasing return period.

From the generated performance curves, the resilience of the buildings infrastructure to the simulated flooding hazards was quantified according to Eq. 8. This resulted in a 300-day resilience assessment of 0.94 for the 5-year event, 0.82 for the 50-year event, and 0.69 for the 500-year event. In practical terms, each value reflects the ability or inability of the system to maintain

its function during the reference period for the given flood event, with zero corresponding to a complete loss of function and one when unaffected. These values and their respective confidence intervals are provided in Table 9.

Because there is an amount of each resource associated with every task, it is therefore possible to produce a timeline of resource utilization for each MCS iteration from the task schedules. Similarly, the MCS iterations were combined to derive the mean and 95% confidence intervals for resource usage. Fig. 4 shows the results of this calculation for cement (R1) usage.

The figure shows the probability density function (PDF) of the Beta-PERT distribution used to apply the uncertainty associated with the resource capacity. Therefore, we can notice that cement usage remains below the available capacity for the 5-year scenario, but increasingly enters the range of the capacity limitation for the 50 and 500-year events, respectively.



**Table 9.** Assessed mean recovery times and mean 300-day resilience values for each of the investigated flooding events. The 95% confidence intervals are provided for each value.

| Event | Recovery [days] | | Resilience, 300-Day [-] | |
|---|---|---|---|---|
| | Mean | 95% Conf. | Mean | 95% Conf. |
| 5-year Flood | 77 | [37, 253] | 0.94 | [0.85, 0.98] |
| 50-year Flood | 165 | [103, 444] | 0.82 | [0.70, 0.91] |
| 500-year Flood | 225 | [155, 537] | 0.69 | [0.55, 0.81] |

We will define recovery completion time $t_{rec}$ as the amount of time required for the system to reach an equilibrium state, or "new normal." Estimation of $t_{rec}$ was carried out by creating an empirical cumulative distribution (CDF) of the scheduled times of the end nodes for each iteration of the MCS as the percentage of recovery completions at time $t$. Through regression, it was determined that the empirical CDF was exponentially distributed. Fitting distributions to the empirical CDFs allowed for estimation of the mean and 95% confidence interval. The results of this regression are presented in Table 9. For the flood with return period of 5-years, the mean recovery time over the 500 MCS iterations was 77 days. The mean recovery time increases with the return period to 165 days and 225 days for the 50-year and 500-year flooding hazards, respectively.

### 4.2 Inundation and Building Damages

Buildings in the study area were manually classified into the three categories according to the outlined methodology. While this classification involves making a certain level of assumptions, all efforts were taken to follow the guidelines laid out by the four indicators. A final classification is displayed in Fig. 5. Generally, it can be observed from the figure that informal buildings are tightly grouped and often located close to the drainage canals. Other building types appear more mixed.

Through testing of rainfall durations, a 60-hour rainfall was found to maximize the peak of the generated hydrograph. However, in order to reduce computation time, a 48-hour duration was instead chosen because the hydrograph peak of this duration was within 2% of the 60-hour value. Sampling the mean flood depth in cells occupied by each building footprint provided the information needed to calculate building damage factors according to Eq. 11 and the corresponding building class. Fig. 4 shows the damage states for each building in the study area due to flooding for the three hazard scenarios. There are significant damages resulting from even the high probability event, especially for buildings close to the drainage canals. Damages increase with increasing return period, with a high number of buildings falling into the D3 classification in the 500-year return period flood.

### 4.3 Standardized Assessment Period

The standardized assessment period presented in this work and applied in the case study, makes it possible for resilience values to be compared across different events (and theoretically between different study areas as well). The sole condition is that the reference period of the assessments remains the same.



**Figure 4.** Results of the modeled impacts, recovery and resource utilization. Each column correspond to the three hazard scenarios modeled: 5-year, 50-year, and 500-year flooding events, respectively. For each scenario, the first row presents the damage states of the buildings immediately following the event phase. The second row visualizes the recovery as the performance of the buildings infrastructure over the assessment period. The third row shows the daily usage of cement (one of the 10 resources included in the model) during the recovery process as well as the distribution of the resource capacity.

The adaptation of the standardized assessment period for extreme events relies on the correctness of the assumption that neglecting the event phase does not significantly affect the resilience metric as long as the recovery phase is significantly longer. In order to test and validate this assumption in the current case study, let us consider the 50-year flooding event, with recovery phase performance shown in Fig. 4. Excluding the event phase, the mean 300-day resilience was quantified as 0.82 (Table 9). According to investigation of the hydrographs produced by the flood model for the 48-hour rainfall, inundation





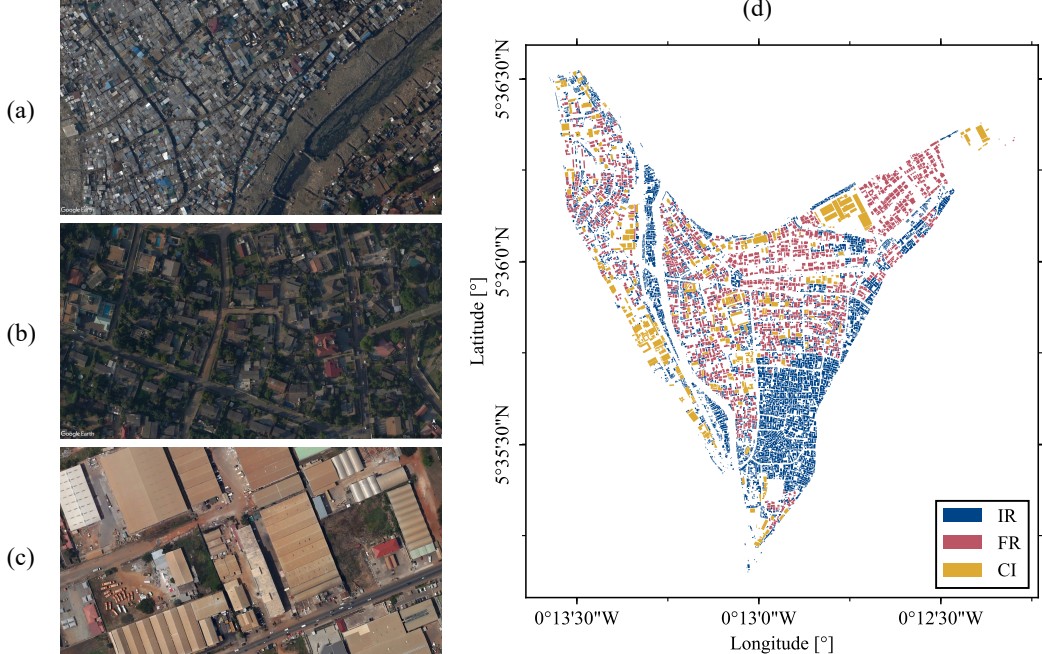

**Figure 5.** Typical areal views of the three building classes (a) IR: Informal Residential [© Google Earth 2023], (b) FR: Formal Residential [© Google Earth 2023], (c) CI: Commercial and Industrial [© Google Earth 2023], and (d) the final building classifications according to the indicators described in the methods section.

lasted approximately 100 hours. A maximum and minimum effect of the event phase can be calculated by assuming that the functionality was either 1.0 or 0.0 throughout the duration of the inundation, respectively. Recalculating the resilience with the addition of the event phase returns a maximum 0.52% difference between including it and excluding it in this case. This

is a relatively insignificant amount given the uncertainty already present in the results. This example adds evidence for the assumption that inclusion of the event phase may not be necessary in cases when the recovery phase is significantly longer than the event phase and quantification of resilience is the primary goal.

## 5   Conclusions

Resilience management has emerged as a potentially more evolved management strategy than that of risk management, the

currently employed standard. In this work, we successfully implement a resilience framework, demonstrating the capabilities of the strategy to produce quantitative estimates of the performance of buildings infrastructure. The methods presented in this work outline a framework for assessing the hazards resilience of infrastructure by modeling recovery as a resource constrained project scheduling problem in a manner that allows for direct comparison between scenarios and potentially across regions and scales.



The results of the case study demonstrate the capabilities of the approach for quantifying the flood resilience of buildings infrastructure. For three flood events with 5, 50 and 500-year return periods, the 300-day resilience of the buildings infrastructure in Alajo was quantified as 0.94, 0.82 and 0.69, respectively. The recovery model also provides an insight into the expected duration of the recovery process. For the 5-year return period event, there was a mean recovery time of the buildings of 77 days, which increased to 165 and 225 days for the the 50-year and 500-year events, respectively. This information is valuable

for identifying the susceptibility of buildings infrastructure to impacts resulting in reduced performance. It is also important information for coordinating responses to flooding events and preparing for the subsequent recovery.

    The presented buildings recovery model relies on the optimization of the RCPSP. Therefore, the largest portion of the computational load is carried by the optimization algorithm. According to the literature, genetic algorithms are an especially capable heuristic approach for estimating the RCPSP (Hartmann and Kolisch, 2000). For this work, we have chosen to apply

the self-adapting genetic algorithm described in Hartmann (2002) because of its performance in comparison with other similar algorithms. Due to the relatively large scale of the damages and the resolution of the tasks composing the recovery scenarios, the use of stratified random sampling as described in the methods was necessary in order to keep the computation time reasonably low with the available hardware. Instead of testing a wide range of optimization algorithms and seeking out additional hardware, this work rather serves to present a framework for modeling recovery based on a novel application of the method.

Many extensions beyond the base RCPSP exist (Hartmann and Briskorn, 2010, 2022) which merit investigation for their applicability for recovery modeling. For instance, all resources are treated as renewable in the case study presented in this work. While that might be an appropriate assumption for some resources (human capital, for example), other resource treatments may be required to better represent the situations being modeled. Future work in the area of recovery modeling using project scheduling methods should explore these additional aspects.

As a strategy for managing the effects of natural hazards on infrastructure, resilience management poses many benefits in comparison with the established practices. While significant work remains before resilience management can be fully operationalized, this approach offers greater insight into the effects of natural hazards on communities beyond the immediate, direct impacts. By focusing on the broader effects, which a resilience-based management strategy considers, managers may discover previously-unknown benefits to applying established interventions and hopefully open the door to new interventions entirely.

*Data availability.* Data from this research are not publicly available. Interested researchers can contact the corresponding author of this article.

*Author contributions.* Conceptualization of the research goals, development of the methodology, and construction of the models was completed by TGJ under the supervision of JL and DKA. TGJ prepared the original draft, which was subsequently reviewed and edited by all co-authors.



*Competing interests.* The authors declare that they have no conflict of interest.



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
