# Peer review of "Quantifying hazards resilience by modeling infrastructure recovery as a resource constrained project scheduling problem"

_EGUsphere, 2023_

## Author Comment (AC1)

As the authors of the manuscript, we would like to begin our response by thanking the anonymous reviewers for their insightful comments and questions. These comments have certainly served to reveal areas in the manuscript which can be strengthened. We hope that you will find the following responses suitable.
* * *
**Reviewer 1, Comment 1:**

The authors describe in line 95ff that agent-based models have been used to model resilience, but have been criticized for their lack of transparency, difficulties to assess and little consensus about model complexity. While the authors argue that their model is an improvement to previous approaches, I struggle to see how the disadvantages of ABMs mentioned do not apply to RCPSP. From reading the manuscript my understanding of RCPSP is that it is very similar to an ABM in terms of specific assumptions of how different building classes are damaged and how specific tasks during the recovery process are carried out. I would argue that this creates the same challenges as using ABM as there is also little consensus about how these tasks are carried out, whether it would be affected by local supply chain issues etc. While I can see the author's point that the model itself is more linear and therefore more transparent, my understanding is that it is still based on a large number of rules and values that are largely based on assumptions or best guesses. Would be great if the authors could discuss how their model and results compare to other approaches modelling resilience.

**Author Response:**

We would like to thank the anonymous reviewer for pointing out this important point. The sentence describing ABM criticisms in Line 95 is not a reflection of our opinion, but instead a summary of the cited articles. We propose the following clarification:

"However, agent-based models are criticized by some for having a lack of transparency, being difficult to evaluate or assess, and needing to strike a balance between being overly simple or overly complicated, for which there is little consensus (Sun et al., 2016; Chen, 2012)."

Regarding comparison with other approaches, we propose to add the following in Results and Discussion:

"Compared with existing models, like those derived from agent-based models (Nejat and Damnjanovic, 2012a; Eid and El-adaway, 2017), the presented model offers relatively more clarity for communicating uncertainty to support decision making. It also simplifies the understanding of specific actions that policy makers, individuals and business owners can undertake to return their damaged buildings to a safe and usable condition following damaging events."

**Reviewer 1, Comment 2:**

It would be great if the authors could say a bit more about how decision makers and planners can use the outcomes of their model. Especially in regard to the 300-day resilience: what does a value of 0.69 mean for planning and decision making? In my view this information is only useful if the outcomes would be presented alongside specific thresholds above which the building is functional/inhabitable again (with restrictions). My impression is that this would be possible to include into the scenarios and could be helpful when planning for shelters, temporary accommodation etc. as it would mark the point when people and businesses can move back.

**Author Response:**

In order to clarify how the model can be used to support planners and decision makers, we propose to add the following text in Results and Discussion:

"A 100-day resilience value of 0.69 is to be interpreted as the infrastructure system providing 69% of its intended performance for the 100 days following the hazard event. In regions were infrastructure is generally more resilient and recovery more expedient, the 100-day resilience may be very high for all events, and therefore the targets can be made more ambitious. For example, managers may set a goal of maintaining 80% functionality for the 10 days following a particular hazard event, or a 10-day resilience of 0.80. By applying this assessment framework, remediation options can be compared in order to select for decisions supporting a particular resilience outcome."

Regarding the suggested application of the model, we propose adding the following text in Results and Discussion:

"Future applications of the recovery model may take advantage of the task schedules produced as output by adding markers for other time dependent actions not considered in this study. For example, it could be possible to monitor the demand on emergency services like shelters by adding tasks which mark the return to occupancy of residential buildings. This was not, however, tested because it was outside the scope of this study."

**Reviewer 1, Comment 3:**

L9ff: I am not sure how useful the numbers are in the abstract as they are not very intuitive to interpret. I would recommend to replace this with a more qualitative statement about what the results mean.

**Author Response:**

In order to make the presentation of the results in the abstract more qualitative, we propose amending the text beginning in Line 9 as follows:

"For the three flood events investigated (5, 50 and 500-year return periods),  the resilience of the buildings infrastructure was successfully quantified and showed a decreasing trend with increasing hazard magnitude. The results of this study are valuable for identifying the vulnerabilities of buildings infrastructure, assessing the impacts resulting in reduced performance, coordinating responses to flooding events, and preparing for the subsequent recovery."

**Reviewer 1, Comment 4:**

L265: How do you come up with 300 days as a suitable assessment period. It seems a bit arbitrary and probably also dependent on the magnitude and size of the event, whether this is a sensible value to present. Was wondering how this metric is more useful than simply reporting the average number of days until full recovery. As mentioned in the general comments I am wondering how discretising the days it takes to recover is useful for decision making.

**Author Response:**

Regarding the selection of 300 days as the assessment period, Section 2.1.2 of the manuscript explains that the assessment period is a constant chosen to envelope the target responses of the system. It is related to the magnitude of the event only to the extent described in the paragraph beginning on Line 174, but should remain constant for the purpose of comparison. The intent is that managers and decision makers should determine an appropriate value for their particular application, but reporting the assessment period along with the metric (e.g. "300-day resilience") is both novel and necessary for understanding the metric.

Concerning the usefulness of the resilience metric compared with reporting the average time to recovery, we propose to add the following text and figure in Results and Discussion:

"The resilience metric applied in this study, quantifies the system performance during the reference period. This provides a more insightful metric compared with other resilience indicators (e.g., time to recovery). Figure X demonstrates the usefulness of the metric by comparing two potential recovery outcomes. Consider that the two scenarios shown in the figure both have the same recovery time $t_{rec}$. However, scenario (a) depicts a more desirable recovery path than scenario (b) because it maintains a higher system performance during the reference period $\Delta t_a$. The resilience metric presented in this study quantifies this difference, whereas assessing recovery time alone does not.

[Figure]

Figure X: Depictions of recovery curves serving as a comparison of two potential recovery outcomes. Although both (a) and (b) have the same recovery time $t_{rec}$, curve (a) depicts a more desirable recovery path than (b) because (a) maintains a higher system performance over the reference period $\Delta t_a$ than (b). The proposed resilience metric reflects this discrepancy and quantifies it ($Re_1 > Re_2$); assessing recovery time alone does not."

**Reviewer 1, Comment 5:**

L284f: Would be good if the authors could mention what the metric for the benchmark performance is.

**Author Response:**

To clarify the metric used to compare the heuristic algorithms, we propose amending the sentence in Line 284 as follows:

"Ultimately, the genetic algorithm described by Hartmann (2002) was chosen due to its benchmark performance compared with other heuristics, in terms of ability to find optimal or best known solutions and the associated computation times when solving standard test sets."
* * *
**Reviewer 1, Comment 6:**

L302: A threshold of 100sqm seems large for footprints of informal buildings. Is there a reason for the threshold being this high?

**Author Response:**

We state in Line 289 that the indicators and their corresponding thresholds are based upon the dataset from the World Bank (2017). While we agree that this could be considered a high value by some, we did not seek to adjust the thresholds utilized by the World Bank in the scope of this work.
* * *
**Reviewer 1, Comment 7:**

L328ff: Would be could to briefly describe how the damage functions were developed and for which region instead of only referring to another paper.

**Author Response:**

To describe the referenced work, we propose amending the sentence in Line 328 as follows:

"The constants corresponding to each building class were derived from regression of the work presented in Englhardt et al. (2019), an analysis of materials-based vulnerability to flooding for buildings in Ethiopia, and are presented in Table 4."

**Reviewer 2, Comment 1**

On line 212 (in assumption 1): what type of tasks are you referring too?

**Author Response:**

In order to make it more clear for readers of the article to understand the meaning of the term "task" we propose to modify the description of Assumption 1 as follows:

"**Assumption 1.** It is assumed that recovery is prolonged by the time required to complete each individual task of the overall process. In this context, tasks are defined as the specific, smaller actions undertaken by the responsible persons or authorities, in progression toward returning the infrastructure to a functional state."
* * *
**Reviewer 2, Comment 2**

On line 403: have you checked the completeness of land-use in OSM? That data is not always fully complete, and may impact your results.

**Author Response:**

To clarify that OSM was not the only source utilized for land-use, we propose to modify the sentence in Line 402 as follows:

"The surface roughness raster was produced by classifying land use into six categories based on a combination of data from the World Bank (2017), OpenStreetMap, and additional manual mapping of relevant features."
* * *
**Reviewer 2, Comment 3**

I would start the results section with section 4.2 (the inundation and damage), and then present the first results of the recovery (now section 4.1). That feels, to me, a bit more logical to read.

**Author Response:**

We agree that a sequential presentation of results is a logical approach. However, we have had success in previous publications by ordering the results according to relevance or importance within the current study. Therefore, we prefer to keep the present ordering.

**Reviewer 2, Comment 4**

Linked to the comment above, I actually miss a discussion sections (also directly the most important comment of my review). The 'results and discussion' section is actually just a 'results' section. As such, I would strongly suggest to include a separate discussion section to actually discuss the results and the methods.

**Reviewer 2, Comment 5 (linked with previous comment)**

I would suggest to include a discussion section that includes the following elements: (i) put the method in perspective with other recovery studies (it is limited of what has been done, but good to discuss how this model has improved the field/is different from others), (ii) put the results in perspective (can we validate some of the results, even if its just anecdotal evidence)? And implications of this work.

**Author Response:**

Thank you very much for this suggestion. Section 4.3 "Standardized Assessment Period" contributes a discussion of one of the most important aspects of the methodology. We have already proposed some discussion points in the author responses to Reviewer 1 (see Author Response to Reviewer 1, Comment 1, 2, and 4). Specifically, the response to Reviewer 1, Comment 1 addresses the recommendation to put the method in perspective with other recovery studies.

Regarding validation of the results, we suggest to add:

"No actual events were simulated in the course of this study. Instead, design storms were applied in the flood model. Therefore, there were no actual flooding impacts to compare to the simulated events, which prevented direct validation of the recovery model results. Future applications of the model may simulate historical events in order to perform a validation. This, however, was beyond the scope of this study."

The implications of the study are discussed in our response to Reviewer 1, Comment 2.

**Reviewer 2, Comment 6**

Finally, one of the most important issues: why is the data and code not available? It is not really of this day and age anymore to not publish the code along-side an article in which a method is being presented. And most of the data is based on open data anyways (e.g. the data extracted from OpenStreetMap)? If the code becomes available, other researchers can also benefit much more of this work. You can just license the work to make sure people use it the way you want them to use it.

**Author Response:**

Thank you very much for this suggestion. We write in the Data Availability statement that, "Interested researchers can contact the corresponding author of this article." While we may decide to make the code openly available in the future, at the present time, we prefer to maintain this policy.

---

## Author Response (AR2)

The authors would like to thank the anonymous referees for taking the time to review our revised submission and for their helpful comments and suggestions. We hope the following responses as well as the updated manuscript adequately address these comments.
* * *
**Reviewer 1, Comment 1:**

(None)

**Author Response:**

Thank you very much for reviewing our manuscript.
* * *
**Reviewer 2, Comment 1:**

Thank you for resubmitting your article. And apologies for the delay. As indicating in the previous round of revisions, I think the paper is well-written and provides a very interesting approach.

However, it would really be very valuable to the academic community if you would provide (at least) a minimal working example of the model you are presenting. It is not really anymore of this day and age to not make code (and data) publicly available. As such, I would ask and urge you to make the code publicily available to the academic community. Reproducibility of your work should be a key aim of each academic.

**Author Response:**

Thank you for pointing this out again. We have now made it explicit in the manuscript by updating the data availability statement:

"Relevant data is available to researchers upon direct request to the corresponding author of this article."
* * *
**Reviewer 2, Comment 2:**

Besides that, it would be nice to elaborate still a bit more on the validation. I understand that you did not model a specific event, but you can still compare your results with (anecdotal) evidence of previous similar events. How do the results compare to similar flood events that have happened in the past?

**Author Response:**

Thank you very much for your comment. We have added the following text in Line 487:

"The magnitude and severity of recently observed floods in the region (Ahadzie et al., 2022; Amoako and Boamah, 2015) are well within the ranges presented in this work."

---

## Author Response (AR3)

**Editor's Note**

Dear Taylor Glen Johnson, Jorge Leandro, and Divine Kwaku Ahadzie,

Thank you very much for providing a revised and updated version of your manuscript.

I am sorry to say that your response to the request for a comparison of your results with past events for validation is not satisfactory. What is meant by the statement "The magnitude and severity of recently observed floods in the region (Ahadzie et al., 2022; Amoako and Boamah, 2015) are well within the ranges presented in this work"?
The magnitude and severity of precipitation amounts or flooded areas? Ranges of impacts?
Looking into the cited papers does not give an understanding. A minimal quantitative statement or specific comparison to previous events can certainly be made.
Please do address this point as a technical correction of your manuscript.

Best regards
Kai Schröter
* * *
**Author Response**

Dear Kai Schröter,

We regret that our previous response did not satisfactorily address the concerns of the reviewer. We have updated the manuscript beginning in Line 487 as follows:

"The near-annual frequency by which flood disasters occurred in recent years (Amoako and Boamah, 2015), provides evidence that the severity of flooding extents and building damages presented in this work are not unprecedented, even in the case of high-probability events. Further, recovery activities like drying of the building components can take months to achieve when specific drying equipment is not available (Ahadzie et al., 2022). This gives us confidence that the intense damages and corresponding lengthy recoveries shown in the results are similar in magnitude to those experienced in reality."

Best regards,
Taylor Glen Johnson, Jorge Leandro, and Divine Kwaku Ahadzie